# COVID-19 Stressors on Migrant Workers in Vietnam: Cumulative Risk Consideration

**DOI:** 10.3390/ijerph18168757

**Published:** 2021-08-19

**Authors:** Ha Thi Thu Bui, Duc Minh Duong, Thanh Quoc Pham, Tolib Mirzoev, Anh Thi My Bui, Quang Ngoc La

**Affiliations:** 1Faculty of Social & Behavioural Sciences, Hanoi University of Public Health, 1A Duc Thang Road, Duc Thang Ward, Bac Tu Liem District, Hanoi 119000, Vietnam; bth@huph.edu.vn; 2Faculty of Fundamental Sciences, Hanoi University of Public Health, 1A Duc Thang Road, Duc Thang Ward, Bac Tu Liem District, Hanoi 119000, Vietnam; pqt@huph.edu.vn (T.Q.P.); lnq@huph.edu.vn (Q.N.L.); 3Nuffield Centre for International Health & Development, Leeds Institute of Health Sciences, 6 Clarendon Way, Woodhouse, Leeds LS2 9NL, UK; tolib.mirzoev@lshtm.ac.uk; 4Health Management Training Institute, Hanoi University of Public Health, 1A Duc Thang Road, Duc Thang Ward, Bac Tu Liem District, Hanoi 119000, Vietnam; btma@huph.edu.vn

**Keywords:** COVID-19, Vietnam, migrant workers, cumulative risk assessment (CRA), industrial zones

## Abstract

This study explored the impact of COVID-19 on migrant workers in Vietnam, using a cumulative risk assessment (CRA) framework which comprises four domains (workplace, environment, individual and community). A cross-sectional study was conducted. Data were collected in 2020 through a self-administered questionnaire with 445 domestic migrant workers in two industrial zones in two northern provinces (Bac Ninh and Ninh Binh) in Vietnam. The majority of migrant workers were female (65.2%), aged between 18 and 29 years old (66.8%), and had high school or higher education level qualifications. Most migrant workers had good knowledge about preventive measures (>90%) and correct practices on COVID-19 prevention (81.1%). Three health risk behaviors were reported: 10% of participants smoked, 25% consumed alcohol and 23.1% were engaged in online gaming. In terms of workplace, occupational working conditions were good. Noise was the most commonly reported hazard (29%). Regarding environment, about two-thirds of migrant workers lived in a small house (<36 m^2^). Most participants (80.4%) lived with their families. About community domain, many reported low salary or losing their job during January–July, 2020. Most migrants received information about COVID-19. The migrant workers suffered from poor health and low occupational safety, fear of job loss and income cut, poor housing and living conditions and limited access to public services. The holistic approach to address stressors is recommended to improve health and safety of migrant workers.

## 1. Introduction

On 11 March 2020, the WHO declared the novel coronavirus disease (COVID-19) as a global pandemic. Since then, countries have implemented measures to address profound social, economic and health consequences, including disruptions of essential social services [1,2]. About 345 million full-time jobs were lost worldwide in the third quarter of 2020 [3]. Governments have taken unprecedented actions to counteract the economic and labor market impacts, including temporary wage subsidies, extending social protection and providing support to keep businesses afloat. However, in spite of these interventions, the crisis is far from over.

Migrant workers are most vulnerable to this pandemic [2,4] as they often: (1) engage in low paid jobs and experience long working hours; (2) work in unsafe working conditions with little occupational safety and health measures; (3) have limited access to public services, such as healthcare and health insurance; (4) have poor housing and living conditions and (5) face cultural and language barriers [5]

Migrant workers account for 4.7% of the total global labor workforce [6]. The principal interventions for reducing COVID-19 transmission for migrants are the same in all contexts, that is, reducing physical contact, improving hygiene and providing financial and non-financial support (e.g., information, equipment, supportive policymaking) [2]. Migrants must be included in national public health systems [7]. Community engagement for health was shown to be effective in dealing with COVID-19 in different countries [8,9]. The migrant workers can become even more vulnerable during the pandemic due to lockdowns and containment activities, threats of job losses, food insecurity, loss of family income and constrained access to effective surveillance early-warning systems and health services. Substantial short and long term mitigation measures are therefore needed to address the needs of migrant workers, particularly in low middle income and low-income countries [10].

The first case of COVID-19 in Vietnam was declared on 23 January 2020. Until 25 April 2021, Vietnam had about 2800 cases with 35 deaths, and the country had successfully contained the third wave of COVID-19 outbreak [1]. Vietnam’s strict containment measures and integration of resources from multiple sectors including health, mass media, transportation, education, public affairs, and defense have reportedly significantly reduced the spread of the epidemic in the country [11].

Since the comprehensive socio-economic reform, known as ‘*Đổi Mới*’ in 1986, domestic migration has significantly increased in Vietnam [12,13]. In 2015, the Vietnam National Internal Migration survey revealed that about 13.6% of population were internal migrants. The rural-urban migration is the most prevalent and internal migration was mostly intra-regional. The migrants are facing barriers to accessing public social services such as health insurance, education for children, reduced electricity rates and programs for poverty reduction [14,15,16]. Poor housing conditions are of concern by most of migrants. Over 50% live in temporary housing or at work sites in unsafe, cramped and unhygienic conditions; 18.4% have an average living space of less than 6m^2^ [15]. Migrant workers in industrial zones experience reproductive health infections [12,13,17], unsafe sex behaviors [18], occupational diseases, work-related injuries and respiratory diseases [19]. However, most migrants have limited access to healthcare services. Migrants have accepted the reality of a “hard life” compromising poor working environment and living conditions in exchange for income and an expectation of a better life afterwards [20]. They mostly prioritize jobs and income, safety and health conditions, but are less involved in community activities [16]. Economic vulnerabilities were found to be the main reason for low utilization of health services among migrants [19].

On 13 April 2020, Vietnam reported the first patient with COVID-19 in Samsung industrial sites in Bac Ninh province of Vietnam [21]. The high risk of transmission of COVID-19 amongst people within the same industrial zones and local communities was becoming a risk of these zones becoming disease clusters [19]. The migrant workers in the industrial zones, one of the most vulnerable population groups, are at higher risk of infecting and developing serious illness from COVID-19. Yet, in Vietnam, little empirical data have been published about the migrant workers’ knowledge and practices on COVID-19 prevention and control.

The cumulative risk assessment (CRA) framework, which was developed by Fox et al. [22] and subsequently adapted by Aladmad for COVID-19 [23], has guided this study (Figure 1). The framework highlights the interactions of multiple stressors in four domains, that can influence health and livelihood of migrants: workplace (occupational hazards); ambient environment (housing and living conditions); individual (socio-economic status, health behaviors, health insurance and practices towards COVID-19 prevention and control); and community (healthcare services, incomes, support towards COVID-19).

The study explored the impact of each stressor on migrant workers during COVID-19 in Vietnam, and suggested potential interventions to improve the health and livelihood of migrant workers in the country across the four domains (workplace, environment, individual and community).

## 2. Materials and Methods

### 2.1. Design

This study applied a cross-sectional study design via online self-administered questionnaire. The data collection was conducted during August–October 2020. The study was funded by the Alliance of Health Policy and System Research, of the World Health Organization.

### 2.2. Study Setting

Two industrial zones (Que Vo and Phuc Son) in two northern provinces in Vietnam (Bac Ninh and Ninh Binh) were purposively selected, where most of workers are migrants.

### 2.3. Sample Size and Sampling

The convenience sampling approach were used to select companies. We calculated the sample size by estimating a population proportion with specified absolute precision (95% confidence level (z), absolute precision at 7%, and maximum population variability (*p* = 0.5)). We, therefore, invited 500 migrant workers from the two industrial zones (Que Vo and Phuc Son). A total of 445 domestic migrant workers participated (89% response rate), including 219 in Que Vo zone and 226 in Phuc Son zone.

### 2.4. Data Collection

Migrant workers aged 18 and over were the main target group for this online survey using a self-administered questionnaire. Most of migrant workers (>99%) in selected companies had an internet connection due to the popularity of cheap smartphones and low 4G fees in Vietnam. A list of all migrant workers from a convenient sample of companies was collected in the two zones. Then, we randomly selected 250 migrant workers from this list. The selected migrant workers were asked to participate via e-mails, text messages or direct phone calls. Prior to the survey, migrant workers who were able to answer the questionnaire were asked to voluntarily participate. Once agreed, the study participants were sent a Redcap link via Zalo or SMS text. The link showed three parts: Part 1: study introduction, Part 2: consent form, and Part 3: questionnaire. After viewing the study introduction, the consent form appeared, and participants could select “No” to stop the survey or select “Yes” to agree to answer the questionnaire. Data was collected during two months (August–October) in 2020.

The questionnaire included information that followed the conceptual framework, including: (1) Individual information (demographic, socio-economic status (SES), employment contract, health and health risk behavior; practice on COVID-19 prevention and control); (2) Workplace: exposure to occupational hazards; job and incomes (3) Environment: housing and living condition, access to clean water, toilet, and internet; and (4) Community engagement: health insurance, access to healthcare facilities, information on COVID-19, supports to migrant workers and economic responsibility

According to the Vietnam Ministry of Health’s guideline on COVID-19 prevention and control, the practice was designed as multiple-choice questions with true-false basis with an additional “I don’t know” option. A correct answer was assigned 1 point and an incorrect/unknown answer was assigned 0 points. Seven preventive measures were included: avoiding crowds, wearing mask, washing hand, avoiding close contact to person with illness, performing physical distance (≥2 m) in public areas, and performing self-report in Bluezone apps.

### 2.5. Data Analysis

Both descriptive and inferential statistics were performed and all independent variables were described under main outcomes by calculating frequencies and percentages. Frequencies of correct answers on practice were described. A comparison between industrial zones according different characteristics was performed (z-test, Chi-square test as appropriate).

### 2.6. Ethical Approval

The study received ethics approval from the Institutional Review Board of the Hanoi University of Public Health (No. 281/2020/HDDD, dated 25 August, 2020). We reimbursed a prepaid phonecard (equivalent to 2 USD) for the participation in our online survey. All workers were contacted by e-mails or messages from their superiors, but they understood that decision for their participation was voluntary. The data were collected following obtaining of informed consent from each participant and the reporting of results protected the participants’ identities.

## 3. Results

### 3.1. The Individual

A total of 445 migrant workers participated in the study. The proportion of female workers was higher than male (65.2% vs. 34.8%); two thirds were young between 18 and 29 years old (66.8%); two thirds had high school and vocational education and one third had college and university level; 41.8% were single; one fourth was from an ethnic minority. Almost all had an employment contract with a company (99.5%) (Table 1).

There were significant differences between both regions in terms of gender, education, ethnicity, marital status and having employment contract (*p* < 0.05) (Table 2). Specifically, the proportion of female workers in Ninh Binh province was significantly higher than in Bac Ninh provinces (70.8% > 59.7%; *p* < 0.05); the proportion of workers with college and university level in Bac Ninh was higher than in Ninh Binh (47.8% > 24.2%; *p* < 0.05); the proportion of workers with ethnic minority origin in Bac Ninh was higher than in Ninh Binh (38.5% > 11%, *p* < 0.05); the proportion of married workers in Bac Ninh was higher than those in Ninh Binh (50.9% > 32.4%, *p* < 0.05); the proportion of workers with employment contract in Ninh Binh was higher than in Bac Ninh (100% > 99.1%, *p* < 0.05).

The majority of migrant workers complied with preventive measures (>90%), though reporting to health centers when having symptoms (fever, cough) was slightly lower (89.4%). However, 81.1% of migrants had the combined correct practices on COVID-19 prevention (having all practices on COVID-19 prevention). The proportion of correct practices in Ninh Binh province was significantly higher than those in Bac Ninh province (87.7% > 74.8%; *p* < 0.05).

### 3.2. Workplace

The occupational working conditions were good according to the worker’s self-reports. Only a few people reported occupational hazards: 2.9% were exposed to high temperature conditions; 4.9% to dusty conditions. However, noise was quite a common problem (29%). The exposure to dust, noise and chemical in Que Vo, Bac Ninh province was significantly higher than in Phuc Son, Ninh Binh (*p* < 0.05) (Table 3).

More than two thirds of workers reported good health. Few health problems related to mobility, self-care, acute and chronic diseases were reported. No significant differences between two provinces (*p* > 0.05) were identified.

The reported health risk behaviors included smoking, drinking and online gaming. The proportion of smoking was about 10%; drinking 25%; and playing online games 23.1%. However, there were significant differences between the provinces. The proportion of migrants who self-reported smoking, drinking and playing online games in Bac Ninh was significantly higher than in Ninh Binh province, respectively (16.8% > 3.7%; 40.3% > 10.5%; 31% > 15.1%; *p* < 0.05).

### 3.3. The Environment

About one third of migrant workers lived in houses with areas of 25–36 m^2^ (31.9%), and 28.1% of migrants lived within 12–4 m^2^. Few people lived in smaller houses of <12 m^2^ (16.2%). The proportion of those living in areas <24 m^2^ was significantly higher in Bac Ninh than in Ninh Binh province (*p* < 0.05) (Table 4).

The majority of people in both provinces were living with families (80.4%). The proportion in Ninh Binh was significantly higher than in Bac Ninh (94.1% > 67.3%, *p* < 0.05). The proportion of migrant living with friends in Bac Ninh was significantly higher than in Ninh Binh (27% > 14.6%, *p* < 0.05).

The majority of migrant workers had access to clean water, toilets and sanitation and internet (>94%). However, the proportion of those with access to clean water and internet in Ninh Binh was significantly higher than in Bac Ninh province (100% > 97.8%; 98.2% > 91.6%; *p* < 0.05).

### 3.4. The Community

Before the COVID-19 pandemic, most workers were in full employment (90.1%) (Table 5). During COVID-19 (January–July, 2020), the proportion of people with full employment was reduced to 86.7%. However, from August to October, the proportion of people with full employment had increased to 96.4%. The proportion of people having unemployment or temporary layoffs was higher in Bac Ninh than in Ninh Binh province, but no significant differences between two provinces were found after COVID-19 (*p* > 0.05).

The average incomes of migrant workers ranged from 6.6 to 7.1 million Vietnamese Dong (equivalent to USD 280–300) per month. The incomes of migrant workers in Bac Ninh were significantly higher than in Ninh Binh (*p* < 0.05).

Most migrant workers (about 70%) were responsible for themselves and their families. The proportion of migrants who had to send remittances to their families was higher in Ninh Binh than in Ninh Binh province (72.1% > 69.5%; *p* > 0.05).

Almost all workers had health insurance (99.6%). About two thirds reported that they could access healthcare services from the hospital and company healthcare units (62.5% and 58.8%, respectively), and lower proportion could receive from local commune health centers (41.4%). Most workers reported that they could seek care at any time of the day (87.2%). However, the proportion of migrant workers, who could receive care from company healthcare unit in Ninh Binh was significantly higher than in Bac Ninh (70.3 > 49.6%, *p* < 0.05) (Table 6).

The proportion of workers, who could receive care at any time was higher in Ninh Binh, while the proportion could receive care during working time was higher in Bac Ninh (*p* < 0.05).

Table 7 provides information that migrant workers received during COVID-19. Over 95% of workers received information about the number of new cases, total cases, number of deaths and information on disease prevention on a daily basis. Proportion of people living in Ninh Binh received information on new case and deaths related to COVID-19 was significantly higher than those in Bac Ninh province, respectively (98.2 > 93.8%; 98.2 > 96.9%; *p* < 0.05).

Sources of information were mainly from the internet (91.7%), text messages via mobile phone (89.2%), television (88.3 %), radio (44%) and leaflets (18.2%). Proportion of people receiving information from TV, radio, internet and leaflets was significantly in Ninh Binh than in Bac Ninh province (*p* < 0.05).

Migrant workers received support from multiple sources. The company was the main sources of personal protective equipment (PPE), food and housing with 54.2%, 41.8% and 37.3%, respectively (Table 8). The support for workers in Bac Ninh were significantly higher than in Ninh Binh province (*p* < 0.05). The other source of supports were community agencies and healthcare facilities. Dormitory landlords had provided lowest support to the workers. Only 1 in 10 migrants received support from landlords.

## 4. Discussion

### 4.1. The Individual

The migrant workers in this study were young, had a good health status, were covered by health insurance, had an employment contract with companies and were highly educated. The characteristics are different from other studies where workers suffered from several illnesses such as limited mobility and occupational diseases [19]. These features could have lowered risks for these migrants towards COVID-19. However, the high presentation of female workers highlights the needs for having sufficient reproductive healthcare services for this group, including health education and annual check-ups [12,17]. The proportion of ethnic minority groups in Bac Ninh is higher than the national average, which suggests a possibility of cultural and language barriers in accessing different public services [24,25]. In terms of religion, almost all migrant workers had no religion, similar to Vietnam in general [15]. In short, female workers, ethnic minority workers and workers with lower education levels deserve specific attention by healthcare units and labor unions in industrial zones in Bac Ninh province.

In the study, the health risk behaviors (smoking, drinking, game online) were 10.3%, 25.6% and 23.1%, respectively. These behaviors could increase the risk of severe COVID-19 illness [22,23]. Cigarette smoking and drinking alcohol are male phenomena in Vietnam, and the data from the study are similar to another study on smoking and drinking behavior among migrant workers [26]. Excessive internet use can constitute an addictive behavior and lead to negative health outcomes, particularly mental disorders [27]. In China, the mental symptoms often associated with health risk behavior during COVID-19 [28]. If COVID-19 is persistent in longer period, the vulnerable groups such as migrant workers will be likely suffer from anxiety, distress and will need support from mental health programs.

Smoking, drinking behaviors and online gaming have very strong associations with the same behavior of peers [27,29]. More than one fourth of migrant workers in Bac Ninh live with friends in dormitories, and, therefore, could be easily influenced by peer behavior. This implies that workplace health promotion or community-based health promotion program on smoking and drinking recession should target groups of workers, particularly those who are young and living with peers.

### 4.2. Workplace

Migrants can be exposed to several occupational hazards from chemical and physical exposures. These could be significantly amplified among workers due to the COVID-19 impact, thus comprising effects of prevention measures such as PPE, safety and health [22,23]. In this study, the exposure to occupational hazardous factors including noise, dust and chemicals was much lower compared to another study in Vietnam [30]. Nevertheless, in order to ensure the occupational health and safety for workers, mitigating impact of occupational factors (noise, dust, chemicals) on workers during COVID-19, the company health unit and labor union should advocate to ensure the compliance with preventive measures for all workers, including PPE, physical distancing and hygiene instructions for all workers.

### 4.3. The Environment

Many migrant workers live in dormitories [15,20] and are exposed to a high risk of overcrowded conditions without access to basic sanitation [15]. In this study, the proportion living with families is high (80.4%) [15] and many have quite good housing and living conditions. Most people have access to clean water, toilets, sanitation and the internet. The good living condition help to reduce the risk of transmission within and amongst households and make it possible to social distance or isolate from family members who may be elderly or have underlying co-morbidities and practice handwashing with soap or hand sanitizers.

### 4.4. The Community

The lives of migrant workers in Vietnam are characterized by insecurity of work and income, poor benefits and limited or no access to government services (health and education services) during the COVID-19 outbreak [31,32]. About 4.6 to 10.3 million workers were affected and sectors that were severely hit included wholesale, retail trade, transport, storage and communication, accommodation and food services and tourism [32]. In this study, the data revealed that during January–July 2020, the strict enforcement of preventive measures such as physical distancing, school closures and travel bans, had an impact on economic activities of industrial zones such as increased layoffs (3.7%) and reduced full-time employment (3.3%). This finding is similar to the global level, where average losses of wages were reported at about 6.5% and were mainly due to reduced working hours and layoffs, and men were less affected than women (4.7% vs. 6.9%) [33].

The migrant workers’ average income level is similar to another study in the same location [20]. The recent study in Vietnam highlighted the critical importance of jobs and job security in personal livelihoods [34]. Individuals with low educational attainment and low incomes are more likely to be exposed to factors contributing to poor health as compared with those with more socio-economic resources [33,34]. Consistent with national level data, in this study the high proportion of migrant workers (over 70%) had responsibility to remit the family [20]. The COVID-19 could result in severe financial hardship and family difficulties if a person could not remit the families due to their job loss or reduced income [22,23]. The stress from the uncertainty of having a permanent job and fear of job loss can also have negative mental health outcomes [22,23].

The more vulnerable groups (such as informal migrant workers without employment contracts, social protection benefits and low educational levels) are more likely to be affected by COVID-19. Many countries introduced wage subsidies to support people during this crisis. Some countries adjusted minimum wages to support low paid workers [33]. The Government of Vietnam also introduced the social assistance package to vulnerable groups, including migrant workers. However, the complex procedures limit the access to this package [35]. Migrant workers could face long-term prospects of poverty and vulnerabilities and increasing inequities compared with other better-off groups [36,37].

This study has shown that the situation improved after the easing of lockdowns, which was very different from other countries and other sectors in Vietnam where migrants continued to suffer from declining working hours, reducing incomes and limited access to social security [32,38]. This could be explained by the companies not belonging to the sectors most severely hit by COVID-19 [39]. Future research could focus on informal migrant workers and the sectors that were more severely hit by the COVID-19, to provide useful inputs into social protection policies.

Lack of access to healthcare could have a severe impact on health if a person contracted the disease [22,23]. The study has shown that the majority of migrant workers could maintain health care access during COVID-19, including hospitals, company healthcare units and commune health stations. This finding is consistent with another study in Vietnam [17]. During COVID-19, the hospital and the company healthcare unit were the most common places for healthcare services. The choice of company healthcare unit in Ninh Binh was significantly higher than in Bac Ninh, which indicates the important role of the company healthcare unit as a gatekeeper. If workers develop symptoms of fever or cough, they will be sent immediately to a company’s health unit for further action: testing, isolation, quarantine or hospitalization. This demonstrates high compliance by these companies with preventive measures. This approach would help to reduce the risk of having self-treated without medication and increasing risk of disease transmission and developing complications [19]. The situation was very different from other countries, where workers had to work due to economic pressures which could further promote transmission of COVID-19 and potentially delay seeking care [40]. This highlights the emergence of having a good company healthcare unit in avoiding the spread of diseases in crowded places, such as industrial zones.

Lack of access to health information on COVID-19 could have resulted in poor compliance with preventive measures and consequences in delayed healthcare if migrant workers counteracted the diseases [22,23]. In this study, the migrant workers received information from different channels. The internet was the most frequent channel, followed by mobile phone SMS and TV. The finding is consistent with another study [20]. The government has utilized wide availability of internet access and mobile phones for providing updated and reliable information on COVID-19, which could contribute to shaping compliant behavior towards COVID-19 prevention and control. The self-health report apps (Bluezone and Ncovi) were developed and people were encouraged to use them, but their uptake was low among the migrant workers in this study [11]. This finding is consistent with another study, where limited use of mobile phones for downloaded apps on health application in Vietnam was reported [41]. In order to effectively utilize the apps for self-report on COVID-19, the government should optimize the content and interfaces of apps that meet the needs of population, to make them more user-friendly and ensure protection of users’ privacy and confidentiality.

During COVID-19, the migrants can suffer from lack of basic PPE, unaffordable costs of food and accommodation, all of which have consequences on health outcomes [22,23]. In this study, migrant workers received different supports (PPE, foods and subsidies for housing) from various sources: companies, dormitory landlords, community agencies, healthcare facilities and family/friends. The company was the most common source, followed by community agency and healthcare facilities. This was different from another study, where families and friends were the most frequent sources of assistance and very few migrants relied on other organizations [15]. Following guidance from the government on supporting vulnerable groups, the companies and organizations provided very good examples of such support. The finding confirms the importance of community engagement in combating with COVID-19 and the needs to strengthen this in the future [8].

The majority of migrant workers complied with preventive measures (81.1%). Our findings suggest that health information on COVID-19 in different channels aimed at improving COVID-19 knowledge can be helpful for maintaining safe practices [42]. Government’s enforcement on preventive measures and the reported sanction of violated cases in TV and other official channels are helpful [42,43]. Therefore, people try to follow the measures to avoid the sanctions. This suggests the importance of continued health risk communication via different official channels to maintain the high awareness and compliance with preventive measures in the community.

### 4.5. Framework

The study adopted a Cumulative Risk Assessment (CRA) from Fox [23] and Aladmad [23]). Although the framework focused on the health impact of COVID-19 on occupational domains, we see the strong influence from other related stressors from different sources across four domains: workplace, environment, community and individual. However, we see clearly the variation of stressors and exposures depending on the specific location. The occupational exposure to COVID-19 does not occur in the isolation from other factors. Understanding influences of all factors is important for effective interventions. No single effective intervention could be effective [23]. The local authorities and employers must address job health and safety needs of migrant workers. The government should have the public policy to support to those affected by COVID-19. The combined efforts must be addressed all the domains of this framework.

This conceptual framework informed reviewing all domains related to migrant workers during COVID-19 in this study, including individual, workplace, environment and community including healthcare access, health communication and community support. A holistic approach to addressing the stressors on COVID-19 should be recommended, including stopping the spread of diseases, ensure essential healthcare services, access to basic health information and community supports.

### 4.6. Limitation

This study has several limitations. Our study was a cross-sectional study; therefore, we were not able to track how the domain characteristics would change over time. Secondly, most information that we acquired from the sample was self-reported information which may subjected to recall bias by the participants. Finally, this study presents case-specific samples in industrial zones in two provinces and the generalization of findings may be limited.

## 5. Conclusions

Migrant workers were particularly vulnerable during the COVID-19 pandemic due to strict preventive measurements, such as lockdown and containment activities. The cumulative risk assessment framework was used to explore the level of exposure and impact of stressors among four domains (individual, workplace, environment and community). The migrant workers could suffer from poor health and low occupational safety, fear of job losses and income cuts, poor housing and living conditions and limited access to public services. However, the levels of stressors and their exposure can vary across the provinces and domains. The holistic and context-specific approach to address stressors is recommended to improve health and safety of migrant workers.

## Figures and Tables

**Figure 1 ijerph-18-08757-f001:**
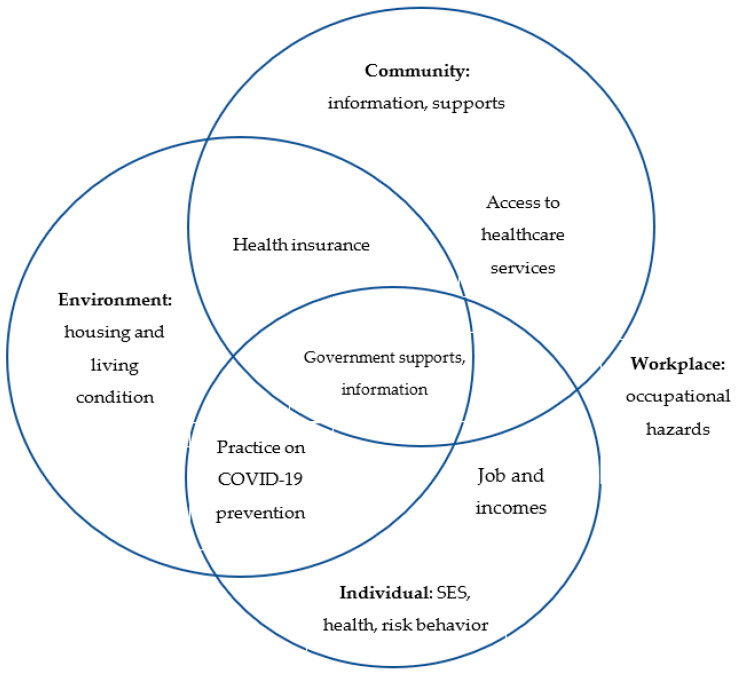
Conceptual Framework in combating COVID-19 for migrant workers. (The framework developed and adapted from Fox [22] and Aladmad [23]).

**Table 1 ijerph-18-08757-t001:** Demographic information of migrant workers.

Variable	Phuc Son- Ninh Binh	Que Vo- Bac Ninh	*p*	Total
	*n* = 219	*n* = 226		*n* = 445
Sex			0.015	
Males	64 (29.2%)	91 (40.3%)		155 (34.8%)
Females	155 (70.8%)	135 (59.7%)		290 (65.2%)
Age groups			0.20	
18–24	62 (28.3%)	78 (34.5%)		140 (31.5%)
25–29	76 (34.7%)	81 (35.8%)		157 (35.3%)
30–34	57 (26.0%)	41 (18.1%)		98 (22.0%)
35–40	17 (7.8%)	22 (9.7%)		39 (8.8%)
41–50	7 (3.2%)	4 (1.8%)		11 (2.5%)
Education levels			<0.001	
Secondary-school	25 (11.4%)	19 (8.4%)		44 (9.9%)
High-school	139 (63.5%)	84 (37.2%)		223 (50.1%)
Vocational	2 (0.9%)	15 (6.6%)		17 (3.8%)
College	24 (11.0%)	43 (19.0%)		67 (15.1%)
University and higher	29 (13.2%)	65 (28.8%)		94 (21.1%)
Religion			0.25	
Buddhism	18 (8.2%)	10 (4.4%)		28 (6.3%)
Catholicism	9 (4.1%)	5 (2.2%)		14 (3.1%)
Christianity	3 (1.4%)	1 (0.4%)		4 (0.9%)
Non-religious	188 (85.8%)	209 (92.5%)		397 (89.2%)
Other	1 (0.5%)	1 (0.4%)		2 (0.4%)
Ethnicity			<0.001	
Dao	0 (0.0%)	9 (4.0%)		9 (2.0%)
Kinh	195 (89.0%)	139 (61.5%)		334 (75.1%)
Muong	22 (10.0%)	14 (6.2%)		36 (8.1%)
Nung	0 (0.0%)	19 (8.4%)		19 (4.3%)
Other	1 (0.5%)	12 (5.3%)		13 (2.9%)
Tay	1 (0.5%)	33 (14.6%)		34 (7.6%)
Marital status			<0.001	
Divorced	5 (2.3%)	5 (2.2%)		10 (2.2%)
Married	143 (65.3%)	106 (46.9%)		249 (56.0%)
Single	71 (32.4%)	115 (50.9%)		186 (41.8%)
Employment contract			0.015	
No	0 (0.0%)	2 (0.9%)		2 (0.4%)
Yes	219 (100.0%)	224 (99.1%)		443 (99.6%)

**Table 2 ijerph-18-08757-t002:** Migrant worker’s practices on COVID-19 prevention and control.

Variable	Phuc Son- Ninh Binh	Que Vo- Bac Ninh	*p*	Total
	*n* = 219	*n* = 226		*n* = 445
Avoid crowded places	217 (99.1%)	222 (98.2%)	0.43	439 (98.7%)
Frequently wash hands	214 (97.7%)	222 (98.2%)	0.70	436 (98.0%)
Wear face mask in the public places	216 (98.6%)	224 (99.1%)	0.63	440 (98.9%)
Avoid contacting people with illness	210 (95.9%)	214 (94.7%)	0.55	424 (95.3%)
Report to health center when having symptoms (fever, cough)	206 (94.1%)	192 (85.0%)	0.002	398 (89.4%)
Keep minimum of 2 m when talking	208 (95.0%)	211 (93.4%)	0.47	419 (94.2%)
Self-report by apps (Bluezone)	201 (91.8%)	206 (91.2%)	0.81	407 (91.5%)
**Correct practice**	**192 (87.7%)**	**169 (74.8%)**	**<0.001**	**361 (81.1%)**

**Table 3 ijerph-18-08757-t003:** Occupational safety and health.

Variable	Phuc Son- Ninh Binh	Que Vo- Bac Ninh	*p*	Total
	*n* = 219	*n* = 226		*n* = 445
Exposure to Occupational Hazard				
Exposed to high temperature	4 (1.8%)	9 (4.0%)	0.18	13 (2.9%)
Exposed to dust *	3 (1.4%)	19 (8.4%)	<0.001	22 (4.9%)
Exposed to noise *	37 (16.9%)	93 (41.2%)	<0.001	130 (29.2%)
Exposed to poor light	3 (1.4%)	3 (1.3%)	0.97	6 (1.3%)
Exposed to humid	2 (0.9%)	4 (1.8%)	0.43	6 (1.3%)
Exposed to chemical *	6 (2.7%)	18 (8.0%)	0.015	24 (5.4%)
Health status			0.30	
Fair	47 (21.5%)	58 (25.7%)		105 (23.6%)
Good	172 (78.5%)	168 (74.3%)		340 (76.4%)
Having chronic disease	4 (1.8%)	6 (2.7%)	0.56	10 (2.2%)
Having acute disease	9 (4.1%)	15 (6.6%)	0.24	24 (5.4%)
Having problem in mobility	4 (1.8%)	8 (3.5%)	0.26	12 (2.7%)
Having problem in self-care	4 (1.8%)	3 (1.3%)	0.67	7 (1.6%)
Health risk behaviour				
Smoking *	8 (3.7%)	38 (16.8%)	<0.001	46 (10.3%)
Drinking *	23 (10.5%)	91 (40.3%)	<0.001	114 (25.6%)
Playing online games *	33 (15.1%)	70 (31.0%)	<0.001	103 (23.1%)

* represents statistically significant at *p*-value <0.05.

**Table 4 ijerph-18-08757-t004:** Housing and living conditions of migrant workers.

Variables	Phuc Son- Ninh Binh	Que Vo- Bac Ninh	*p*	Total
	*n* = 219	*n* = 226		*n* = 445
Residence area *			<0.001	
<12	23 (10.5%)	49 (21.7%)		72 (16.2%)
12–24	50 (22.8%)	75 (33.2%)		125 (28.1%)
25-36	98 (44.7%)	44 (19.5%)		142 (31.9%)
>36	48 (21.9%)	58 (25.7%)		106 (23.8%)
Living with family *	206 (94.1%)	152 (67.3%)	<0.001	358 (80.4%)
Living with friends *	32 (14.6%)	61 (27.0%)	0.001	93 (20.9%)
Having clean water *	219 (100.0%)	221 (97.8%)	0.027	440 (98.9%)
Having toilets and sanitation *	219 (100.0%)	226 (100.0%)		445 (100.0%)
Access to internet at home *	215 (98.2%)	207 (91.6%)	0.002	422 (94.8%)

* represents statistically significant at *p*-value <0.05.

**Table 5 ijerph-18-08757-t005:** Migrant workers’ employment, salary and economic responsibilities.

Variables	Phuc Son- Ninh Binh	Que Vo- Bac Ninh	*p*	Total
	*n* = 219	*n* = 226		*n* = 445
Employment status				
Before COVID-19 *			<0.001	
Full-time employment	211 (96.3%)	190 (84.1%)		401 (90.1%)
Temporarily layoffs	2 (0.9%)	23 (10.2%)		25 (5.6%)
Unemployment	6 (2.7%)	13 (5.8%)		19 (4.3%)
1 January–23 July			0.23	
Full time employment	196 (89.5%)	190 (84.1%)		386 (86.7%)
Temporarily layoffs	14 (6.4%)	23 (10.2%)		37 (8.3%)
Unemployment	9 (4.1%)	13 (5.8%)		22 (4.9%)
23 July–11 October			0.11	
Full time employment	214 (97.7%)	215 (95.1%)		429 (96.4%)
Temporarily layoffs	1 (0.5%)	7 (3.1%)		8 (1.8%)
Unemployment	4 (1.8%)	4 (1.8%)		8 (1.8%)
Income (million VND) (Mean/SD)				
Before COVID-19 *	5.7 (1.8)	7.9 (4.4)	<0.001	6.8 (3.6)
1 January–23 July *	5.3 (1.7)	7.9 (4.4)	<0.001	6.6 (3.6)
23 July–11 October *	5.6 (1.5)	8.5 (4.0)	<0.001	7.1 (3.3)
Economic responsibility				
Worker only *	128 (58.4%)	187 (82.7%)	<0.001	315 (70.8%)
Family remittances	158 (72.1%)	157 (69.5%)	0.53	315 (70.8%)
Other duties	60 (27.4%)	69 (30.5%)	0.47	129 (29.0%)

* represents statistically significant at *p*-value <0.05.

**Table 6 ijerph-18-08757-t006:** Health care support to migrant workers during COVID-19.

Variables	Phuc Son- Ninh Binh	Que Vo- Bac Ninh	*p*	Total
	*n* = 219	*n* = 226		*n* = 445
Healthcare facilities				
Having health insurance	218 (99.5%)	225 (99.6%)	0.98	443 (99.6%)
Hospital *	99 (45.2%)	179 (79.2%)	<0.001	278 (62.5%)
Healthcare unit in company *	154 (70.3%)	112 (49.6%)	<0.001	266 (59.8%)
Commune health center *	62 (28.3%)	121 (53.5%)	<0.001	183 (41.1%)
Time available				
All the time *	203 (92.7%)	185 (81.9%)	<0.001	388 (87.2%)
Working time *	13 (5.9%)	26 (11.5%)	0.038	39 (8.8%)

* represents statistically significant at *p*-value <0.05.

**Table 7 ijerph-18-08757-t007:** Information received on COVID-19.

Variables	Phuc Son- Ninh Binh	Que Vo- Bac Ninh	*p*	Total
	*n* = 219	*n* = 226		*n* = 445
Updated information on COVID-19				
Number of COVID-19 new cases *	215 (98.2%)	212 (93.8%)	0.019	427 (96.0%)
Number of deaths due to COVID-19 *	215 (98.2%)	210 (92.9%)	0.007	425 (95.5%)
COVID-19 prevention	215 (98.2%)	219 (96.9%)	0.39	434 (97.5%)
Source of information on COVID-19				
Television *	210 (95.9%)	183 (81.0%)	<0.001	393 (88.3%)
Radio *	120 (54.8%)	76 (33.6%)	<0.001	196 (44.0%)
Internet *	210 (95.9%)	198 (87.6%)	0.002	408 (91.7%)
Mobile phone message	206 (94.1%)	191 (84.5%)	0.062	397 (89.2%)
Leaflets *	47 (21.5%)	34 (15.0%)	0.001	81 (18.2%)

* represents statistically significant at *p*-value <0.05.

**Table 8 ijerph-18-08757-t008:** Support given to migrant workers during COVID-19 from various sources.

Sources of Supports	Phuc Son- Ninh Binh	Que Vo- Bac Ninh	*p*	Total
	*n* = 219	*n* = 226		*n* = 445
PPE				
Company *	95 (43.4%)	146 (64.6%)	<0.001	241 (54.2%)
Dormitory landlords *	11 (5.0%)	32 (14.2%)	0.001	43 (9.7%)
Community agencies	47 (21.5%)	64 (28.3%)	0.095	111 (24.9%)
Healthcare facilities *	55 (25.1%)	79 (35.0%)	0.024	134 (30.1%)
Family/Friends	38 (17.4%)	46 (20.4%)	0.42	84 (18.9%)
Food				
Company *	77 (35.2%)	109 (48.2%)	0.005	186 (41.8%)
Dormitory landlords *	12 (5.5%)	32 (14.2%)	0.002	44 (9.9%)
Community agencies *	47 (21.5%)	69 (30.5%)	0.029	116 (26.1%)
Healthcare facilities	43 (19.6%)	51 (22.6%)	0.45	94 (21.1%)
Family/Friends *	53 (24.2%)	78 (34.5%)	0.017	131 (29.4%)
Subsides for housing				
Company	82 (37.4%)	84 (37.2%)	0.95	166 (37.3%)
Dormitory landlords *	20 (9.1%)	53 (23.5%)	<0.001	73 (16.4%)
Community agencies	38 (17.4%)	54 (23.9%)	0.088	92 (20.7%)
Healthcare facilities	35 (16.0%)	38 (16.8%)	0.81	73 (16.4%)
Family/Friends	49 (22.4%)	63 (27.9%)	0.18	112 (25.2%)

* represents statistically significant at *p*-value <0.05.

## Data Availability

The data presented in this study are available on request from the corresponding author. The data are not publicly available due to privacy.

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
