# Peer review of "COVID-19 Stressors on Migrant Workers in Vietnam: Cumulative Risk Consideration"

_ijerph, 2021, doi:10.3390/ijerph18168757_

Round 1
Reviewer 1 Report
Abstract : indicate ratio of male/female workers
Ethical : did you pay the migrant workers to ans et. Did you approach them through their companies so that they could feel forced to answer ?
Line 46 : remove « are »
Line 48 : add « have » after 3)
Line 49 : add « have » after 4)
Line 72 : correct the sentence
Section 2.1 : the section needs to be reformulated
Section Data collection : provide more information on how the participants were recrutée. And how long was the data collection period ?
Line 144 : confusion between the two industrial zones names and the two Viernam’s provinces names
Line 186 : two thirds were…
Table 1 : There is a significant difference between both regions in terms of migrant workers’ ethnicity
Table 1 : a vast majority of the sample is non religious, Howard can you interprete this ? Does it have an impact on your result ?
Author Response
Thanks for your comments. Please see our feedback in the attached file

Reviewer 2 Report
This article aims to assess the main impacts of COVID-19 on migrant workers in 19 Vietnam by conducting a cross-sectional study and using four domains of a cumulative risk assessment (CRA) framework. Authors assert a need for this study by explaining that migrant workers are most vulnerable to the pandemic due such factors as unsafe working conditions and limited access to healthcare services, and that migrant workers in Vietnam are at higher risk of suffering from severe COVID-19 conditions. This is an insightful paper. Feedback is as follows:
- The authors should further establish the rationale for the study by discussing if there are existing studies that explore this topic (impacts of COVID-19 on migrant workers in 19 Vietnam) or if they are addressing a gap in the literature by assessing this topic. Overall, are the authors adding to existing literature or addressing a literature gap and need for more studies on this topic?
- Lines 62-64- Authors state “Up to now Vietnam has about 1400 cases with 35 death, and the country in the second wave of COVID-19 transmission”. Are these the total cases and deaths since the start of the pandemic? These numbers seem low.
- For lines 68-70 – Authors state “Since socio-economic reform ‘Doi Moi’ in 1986, the enhanced investment and development has resulted in increased internal migration as people have and continue to move away from their communities of origin in search of economic opportunities”. What were the Doi Moi reforms? This should be further explained for the lay reader that may not be familiar with these reforms.
- Line 144 – Regarding sample size and sampling, if 445 domestic migrant workers participated, how many were recruited overall? What was the response rate?
- Lines 144-145- Regarding subject recruitment, since potential participants were migrant workers, were efforts made to minimize potential barriers to participation such as transportation, health literacy, or possible language barriers? Were participants provided incentives to participate or other items (e.g., refreshments) to recognize their time and effort?
- Lines 150-152 – Since “Participants were required to have an internet connection, to voluntarily participate in an online questionnaire, and to be able to read, understand, and answer the provided questions”, did requirement to have internet access pose a barrier for the migrant workers?
- Line 163 – In “access to clean water, WC, internet”, what is meant by ‘WC’?
- Line 202 – In Table 2, what is considered “Correct practice”?
In addition, the manuscript should be reviewed for English language and style.
Overall, this is an insightful, pertinent, and unique study. The paper is interesting to read. Attending to some clarifying questions can help to improve the quality of the paper.
Author Response

(The authors gave the same response as above.)

Round 2
Reviewer 2 Report
The authors have effectively addressed the reviewer feedback. The manuscript is clearer and more cogent and logical. The paper appears suitable for publication.